# Parabrachial *Calca* neurons mediate second-order conditioning

Sekun Park [1,2], Anqi Zhu [1,2], Feng Cao [1,2] & Richard D. Palmiter [1,2,3] ✉

Learning to associate cues, both directly and indirectly, with biologically significant events is essential for survival. Second-order conditioning (SOC) involves forming an association between a previously reinforced conditioned stimulus (CS1) and a new conditioned stimulus (CS2) without the presence of an unconditioned stimulus (US). The neural substrates mediating SOC, however, remain unclear. Parabrachial *Calca* neurons, which react to the noxious US, also respond to a CS after pairing with a US, suggesting that *Calca* neurons mediate SOC. We established an aversive SOC behavioral paradigm in mice and monitored *Calca* neuron activity via single-cell calcium imaging during conditioning and subsequent recall phases. These neurons were activated by both CS1 and CS2 after SOC. Chemogenetically inhibiting *Calca* neurons during CS1-CS2 pairing attenuated SOC. Thus, reactivation of the US pathway by a learned CS plays an important role in forming the association between the old and a new CS, promoting the formation of second-order memories.

Remembering salient stimuli in the surrounding environment is crucial for predicting potential benefits or danger, impacting an animal's survival. Pavlovian conditioning has been extensively studied to understand underlying neural substrates and learning mechanisms[1–8]. It has been widely accepted because of the simplicity of CS and US, rapid learning, robust conditioned responses, and universality across species[9]. The dynamic and complex nature of environments rewards animals that use all available information for more precise prediction[10,11]. Responding to the earliest cue that predicts an outcome is an advantageous strategy and resembles the higher-order learning commonly observed in pets when learning to respond to cues that predict the availability of treats.

While Pavlov is credited with a rigorous description of first-order conditioning (FOC), he also recognized the importance of mechanisms that allow multiple cues to predict outcomes[12,13]. This type of higher-order conditioning includes sensory preconditioning and second-order conditioning (SOC). The preexposure to paired neutral stimuli (CS1 and CS2) prior to any US exposure permits subsequent CS1-US association to be transferred from one stimulus to another in sensory preconditioning[14]. In contrast, with SOC, the CS2 is associated with CS1 after the FOC (CS1 paired with US)[15]. While SOC has been established in various organisms, including *Drosophila*, rats, and humans[15–17], it has

surprisingly not been verified in mice, despite the availability of various genetic models that can lead to cellular and molecular mechanisms of SOC. Studies using rats demonstrated that infusions of NMDA antagonists or muscimol into the basolateral amygdala (BLA) interfere with the acquisition of SOC[18,19]. The activity within the BLA is pivotal for SOC consolidation as evidenced by the impact of post-training protein synthesis inhibition[20,21] or local anesthesia within the BLA[22]. Furthermore, impairing BLA function hinders the extinction of second-order fear memory[19,23]. Additional research highlights the involvement of dopamine and opioid systems in SOC[24,25].

Sensory preconditioning and SOC serve as suitable paradigms for testing complex learning because they are also prevalent in natural environments[13,17]. In this study, we focused on SOC because CS1 already carries learned value when paired with CS2 thereby providing more salience to associate CS2 with CS1, promoting a CS2-induced conditioned response even though CS2 was never paired directly with US. The first-order memory networks are presumably engaged to propagate the value (aversive or appetitive) to CS2 during SOC[10]. Despite the well-known reactivation of neural networks upon re-exposure to learned CS[26–28], the role of US pathways in SOC remains unknown.

The parabrachial nucleus (PBN) in the dorsal pons receives sensory signals from the periphery and projects to multiple forebrain

[1]Howard Hugues Medical Institute, University of Washington, Seattle, WA, USA. [2]Department of Biochemistry, University of Washington, Seattle, WA, USA. [3]Department of Genome Science, University of Washington, Seattle, WA, USA. ✉e-mail: palmiter@uw.edu

regions, including the central nucleus of the amygdala (CeA)[29–33], which is a primary outlet of the amygdala transmitting fear-related information[34,35]. A subset of PBN neurons expressing the *Calca* gene, encoding calcitonin-gene-related peptide (CGRP[PBN] neurons), constitutes about 15% of the neurons in the PBN and is primarily located in the external lateral division of the PBN[36,37]. CGRP[PBN] neurons are excitatory and release glutamate, along with several neuropeptides on to forebrain postsynaptic targets[36,37] and receive inputs from many brain regions[38,39].

CGRP[PBN] neurons are activated by a wide range of aversive and painful stimuli such as foot shock, visceral malaise, and noxious olfactory, gustatory, visual, or auditory stimuli[33,36,39–46]. Due to this broad tuning to aversive stimuli, these neurons serve as a general alarm for alerting animals to potential danger[36,39,46]. Activation of CGRP[PBN] neurons via chemogenetic or optogenetic tools evokes a range of aversive unconditioned responses, such as freezing, anorexia, bradycardia, tactile sensitivity (allodynia) and signs of anxiety[41–45,47]. Transient or permanent silencing of these neurons during FOC attenuates conditioned responses, highlighting their crucial role in the acquisition and expression of aversive, first-order memories[43–45,48]. Notably, CGRP[PBN] neurons become activated by a tone CS previously paired with a foot shock US, or by a novel taste CS paired with visceral malaise US[43,46].

The reactivation of CGRP[PBN] neurons by a CS is comparable to the well-known activation of dopamine neurons by the CS after appetitive learning with a CS-reward pairing[49–51]. Considering that CGRP[PBN] neurons normally serve to relay the aversive US, their reactivation by CS1 could serve as a surrogate US when paired with a new CS2. The appearance of CGRP[PBN] neuronal activity after learning provided the incentive for studying the role of CGRP[PBN] neurons in SOC. Therefore, we aimed to investigate the potential role of CGRP[PBN] neurons in SOC by developing a behavioral paradigm in mice and using single-cell, calcium imaging of CGRP[PBN] neurons to monitor dynamic changes in their activity during SOC. We also traced the activity of the same individual neurons over the course of the study. Additionally, to ascertain the critical role of CGRP[PBN] neurons in SOC, we chemogenetically inhibited these neurons during CS2-CS1 association phase.

## Results

### Development of a second-order conditioning paradigm for mice

We implemented a second-order, fear-conditioning protocol, consisting of four sequential phases across successive days: habituation, first-order association (CS1-US pairing), second-order association (CS2-CS1 pairing), and testing for the conditioned freezing response to CS2 (Fig. 1a). On the first day of the experiment (Fig. 1a), mice were habituated to 10 light cues (first-order CS1) and 10 tone cues (second-order CS2) in random order. The next day, the mice received 10 trials of light CS1 (10 s, 60 lux) that co-terminated with a foot shock US (0.5 mA, 0.5 s); we measured freezing behavior in response to the light CS1 to document learning (Fig. 1b). On day 3, the mice were exposed four times to tone CS2 (10 s, 70 dB) followed 0.5 s later by the light CS1 to establish a second-order association. The control group (receiving only CS1) did not undergo CS2-CS1 pairing, serving as a baseline for comparison. The number of CS2-CS1 association trials was based on prior rat studies and limited to minimize CS1 extinction[10,19]. Both control and experiment groups exhibited moderate freezing in the chamber without any additional cues, likely due to the contextual fear associated with the chamber from day 2 (Supplementary Fig. 1a). Freezing to the light CS1 increased during the CS2-CS1 pairing reflecting their memory of the association with the US (Fig. 1c). On day 4, the test phase, the mice were exposed to the tone CS2 three times (120 s apart) in a novel context to test second-order memory (Fig. 1a). The mice exhibited significant freezing levels to the tone CS2 compared to the control group (Fig. 1d). The percentage of time freezing to the tone CS2 was lower than to the light CS1 on the CS2-CS1 pairing

phase. Specifically, the comparison of second-order memory strength to first-order memory averaged 66.4% across three experiments involving 15 mice (Supplementary Fig. 1b).

Given that FOC to a specific CS not only elicits conditioned responses directly to that CS but also shows stimulus generalization—where conditioned responses extend to similar stimuli[52,53]—we explored this phenomenon in the context of SOC. After establishing a second-order fear memory for CS2 (a 5-kHz tone, as used in our initial experiment), mice were presented with either the novel 10-kHz or the 5-kHz tone CS2 on the test day. Mice showed less freezing to the 10-kHz tone than the 5-kHz tone, indicating that mice could distinguish between these tones after SOC (Fig. 1e). This result suggests that the second-order memory is not generalized to similar types of stimuli under these experimental conditions.

Considering that the mice were exposed to the same context (metal chamber with an electric grid) during both first- and second-order associations, it is possible that the context itself might contribute to the second-order association, either instead of or in addition to the light CS1 because the context CS is capable of evoking contextual fear responses[54]. Therefore, the mice might freeze in response to the tone CS2 during test day due to developing an association between the context and the tone CS2, rather than between the tone CS2 and the light CS1. To test this possibility, we performed an experiment by using the metal chamber only during CS1-US pairing phase and conducting CS2-CS1 pairing in a different context. We observed that the mice still displayed a high level of freezing to the tone CS2 in this modified protocol, which indicates that tone CS2 evokes a freezing response through the association with the light CS1 (Fig. 1f, control and experiment).

In FOC, the initially neutral CS1 becomes a predictive signal of the US after learning[55]. Therefore, we hypothesized that the second-order association is formed only when CS2 precedes CS1. To investigate the impact of the timing of CS2 presentation during second-order association, we introduced CS2 in various temporal configurations. In this experiment, different groups of mice were exposed to simultaneous presentation, reversed order of presentation, or with a 30-s interval between the tone CS2 and light CS1 rather than 0.5 s (Fig. 1f). All these groups of mice failed to acquire a fear memory to the tone CS2, in contrast to the experimental group that received the tone CS2 followed by the light with a 0.5-s interval (Fig. 1f). Taken together, these results reveal that mice efficiently associate the fear response to CS2 when CS2 precedes CS1 with a short interval between them.

To further optimize SOC paradigm, we asked whether repeating first- and second-order associations could enhance conditioned responses to CS2. A group of mice underwent the CS1-US and CS2-CS1 pairing phases as usual, then this sequence was repeated on the next 2 days. After that, they received the tone CS2 to examine second-order fear memory Fig. 1g. Freezing time in mice with two conditioning trials was not significantly different than the mice with only one (Fig. 1g).

Given the importance of memory consolidation in the development of long-term memory in associative learning[56–58], we also tested whether the second-order memory relies on a consolidation period. All mice received the same habitation, followed by different training protocols: the standard 3-day protocol, a 2-day protocol with CS2-CS1 pairing and testing occurring 1 h later, or a 1-day protocol with US-CS1, CS2-CS1 and testing all conducted on the same day (Fig. 1h). Interestingly, all groups exhibited similar freezing to light CS1 (Supplementary Fig. 1c), but only mice that received the 3-day training protocol demonstrated a significant second-order memory compared to the other groups (Fig. 1i). These findings indicate that it takes time (overnight in this example) to effectively form second-order memories in our SOC paradigm.

We posited that second-order memories might extinguish more rapidly than first-order memories, reflecting their relatively weaker

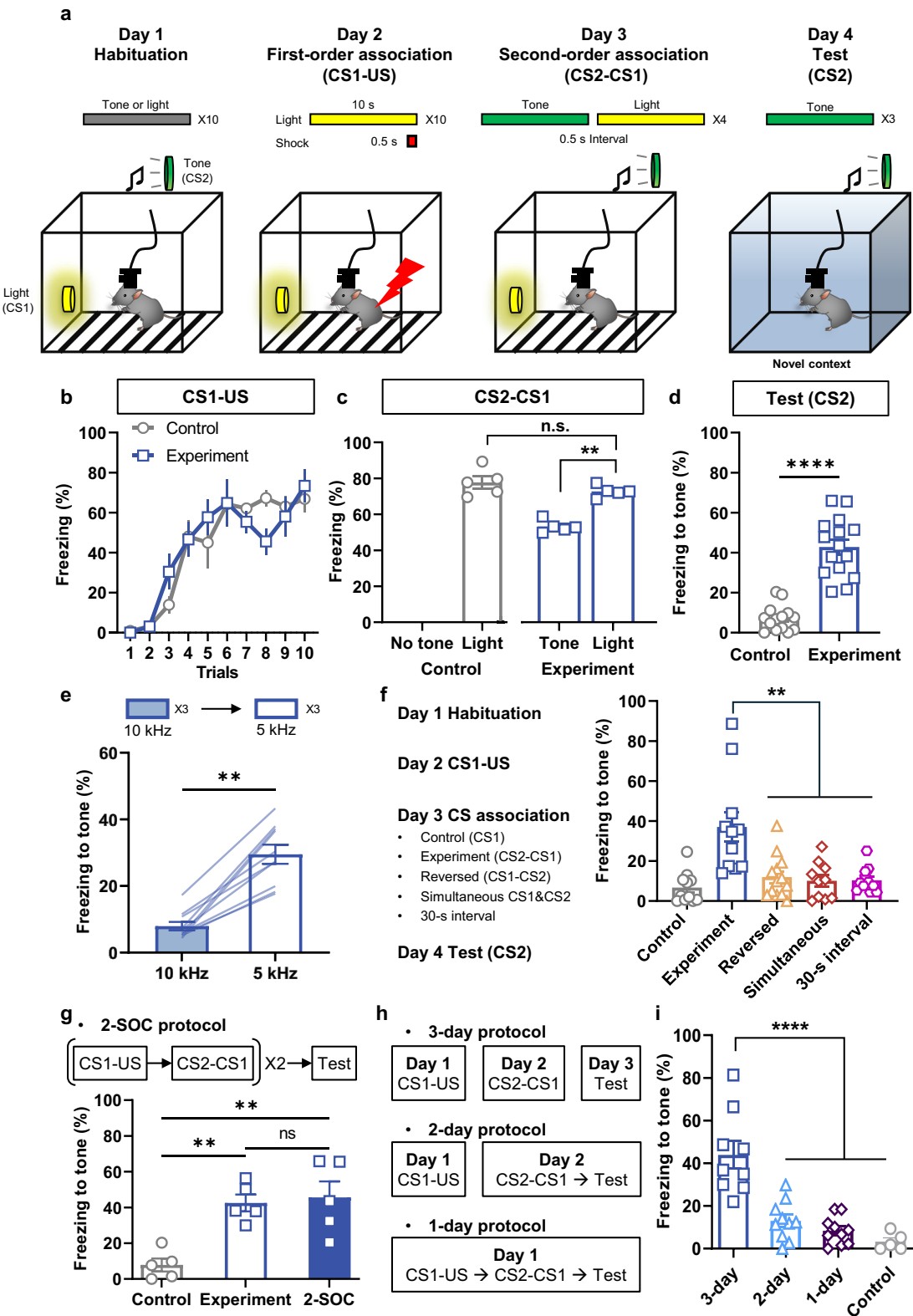

conditioned responses. To test this, mice were exposed to an extended series of 10 light CS1 and tone CS2 trials in FOC and SOC groups, respectively. The mice extinguished their freezing responses to the tone CS2 faster than to the light CS1 (Supplementary Fig. 1d), a result consistent with the slow extinction of the first-order fear memory in previous studies[46].

Sensory preconditioning has a similar experimental procedure to SOC but the CS2-CS1 pairing occurs before the CS1-US pairing. We

habituated mice to light CS1 and tone CS2. On day 2, mice received 4 CS2-CS1 pairings with a 0.5-s interval between CS2 offset and CS1 onset. The next day, there were 10 CS1-US pairings. On day 4, we tested their freezing response to light CS1 and tone CS2; the mice received 3 trials of tone CS2 then 3 trials of light CS1. A control group did not receive the CS2 on day 2. Both the sensory preconditioning and control groups had similar learning curves (Supplementary Fig. 1e). Both groups showed a high freezing level to light CS1 on the test day;

**Fig. 1 | Validation of second-order fear conditioning (SOC) paradigm and behavioral responses in mice. a** Schematic representation of the standard SOC paradigm. **b** Acquisition of first-order fear conditioning measured by freezing behavior to a light CS1 across trials for both control and experimental groups ($n = 5$ for both group). **c** Freezing behavior during the CS2-CS1 association phase. Note that the control group was not exposed to tone CS2 in this session ($n = 5$ for both groups). **d** Average freezing level across 3 trials of tone exposure on the test day ($n = 14$ for control and $n = 15$ for experiment). **e** Freezing levels in response to 10 kHz and 5 kHz tones on the test day ($n = 10$). Note that 5 kHz tone was paired with light CS1, whereas 10 kHz tone was not. **f** Experimental procedure for variable CS2-CS1 associations (Left). Freezing responses to tone CS2 on the test phase for the control, experimental group, and the three timing variants (Right, $n = 11$ for control, $n = 11$ for experiment, $n = 13$ for reversed group, $n = 10$ for simultaneous, $n = 12$ for 30-s interval). **g** Comparison of freezing behavior with one or two training sessions; data from panel (**d**) are included for easier comparison ($n = 5$ for each group). **h** Schematic depiction of SOC protocols with varying training durations. **i** Freezing responses to tone CS2 on the test day across different SOC groups with varied training durations ($n = 9$ for 3-day group, and $n = 10$ for 2-day and 1-day groups and $n = 5$ for control group). Data are presented as mean ± SEM, **$p < 0.01$, ***$p < 0.001$, ****$p < 0.0001$. Source data are provided as a Source Data file.

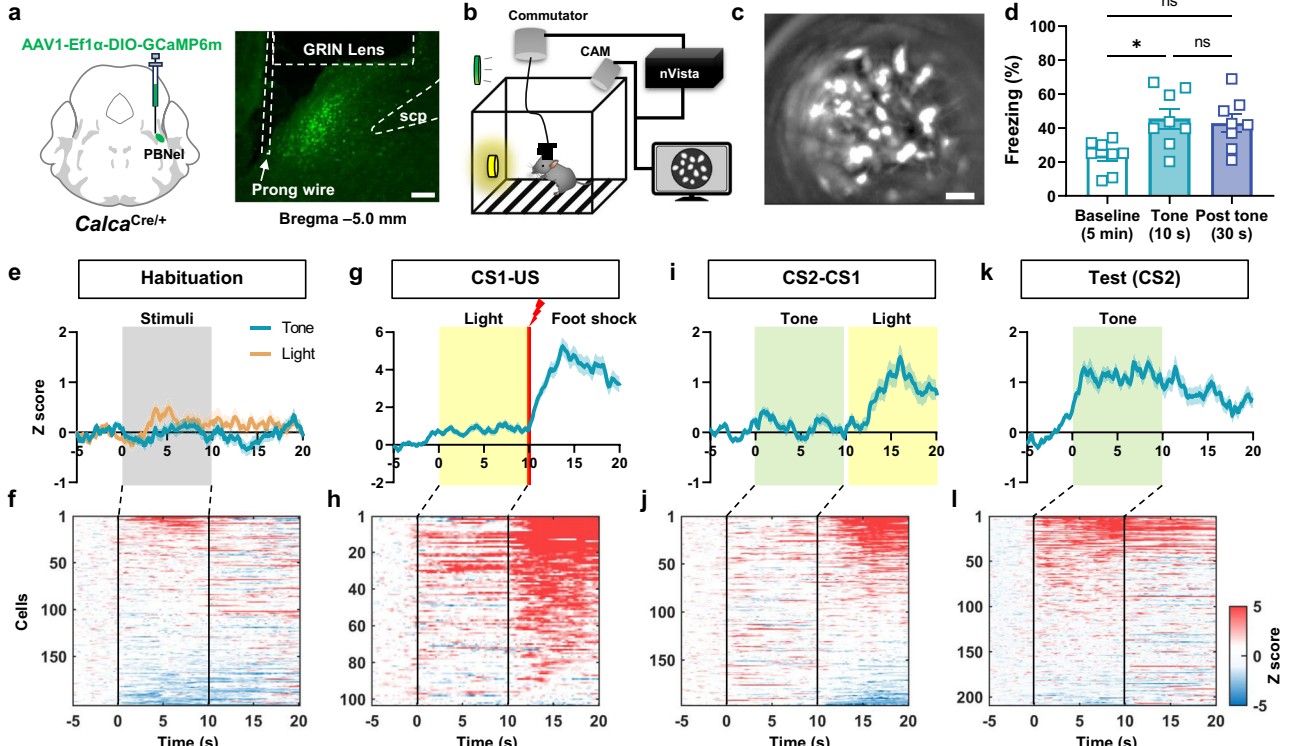

**Fig. 2 | CGRP$^{PBN}$ neuronal activity during second-order conditioning (SOC). a** Schematic illustration AAV-DIO-GCaMP6m viral injection into PBNel (external lateral region of PBN) and a representative image of the GRIN lens and stabilizing prong implanted in the PBN. Scale bar, 100 μm. **b** Schematic illustration of the experimental setup for Ca²⁺ imaging in freely behaving mice. **c** Representative field of view through the GRIN lens displays CGRP$^{PBN}$ neurons expressing GCaMP6m. Scale bar, 50 μm. **d** Freezing responses of animals subjected to Ca²⁺ imaging to the tone CS2 after SOC on test day ($n = 8$). **e** Average fluorescence activity of all neurons during habituation in response to light CS1 and tone CS2 (199 neurons). **f** Individual neuronal responses to tone CS2 that are aligned by AUC during CS2. **g** Average traces of all neurons during the 10 CS1-US pairings (103 neurons). **h** Individual neuronal activity aligned by AUC after foot shock US. **i** Average fluorescence activity of all neurons during CS2-CS1 pairing (197 neurons). **j** Individual neuronal responses aligned by AUC of light CS1 during 4 pairings. **k** Average traces of all CGRP$^{PBN}$ neurons during the test trials (209 neurons). **l** Individual neuronal activity aligned by AUC during tone CS2. Neuronal activity traces are the average of 3 trials. Data are mean ± SEM, *$p < 0.05$. Source data are provided as a Source Data file.

however, the response to tone CS2 was not different from the control group (Supplementary Fig. 1f, g). These results indicate that while the formation of first-order memory was successful, sensory pre-conditioning failed to materialize, likely because the four CS2-CS1 paired trials were insufficient to establish a robust CS2-CS1 association necessary for CS2 to elicit a fear memory through its connection to the CS1-US association.

### Imaging neuronal activity of CGRP$^{PBN}$ neurons during second-order conditioning

We hypothesized that CGRP$^{PBN}$ neurons would be activated by the tone CS2 after SOC like their reactivation by the light CS1 after FOC[46]. To observe CGRP$^{PBN}$ neuronal activity across different phases of the SOC protocol, we used calcium imaging in freely behaving mice by injecting AAV-DIO-GCaMP6m (a proxy indicator of neuronal activity[59]) into the PBN of *Calca*$^{Cre/+}$ mice and implanting a GRIN lens over the external lateral region of the PBN (Fig. 2a, b and Supplementary Fig. 2a, b). After allowing 4-6 weeks for GCaMP6 expression and stabilization of the imaging field (Fig. 2c), the mice were habituated to the light CS1 and tone CS2. Variability in the number of neurons imaged each day was due to motion and changes in the field of view caused by reattaching the microscope. Eight mice (both sexes) were included in the analysis, and they exhibited significant freezing to 10-s tone CS2 on the test day (Fig. 2d). During habituation (day 1), the average fluorescence of CGRP$^{PBN}$ neurons (199 neurons imaged) did not change to the tone or light (Fig. 2e, f). On day 2, the mice were presented with 10 trials of a light CS1 that co-terminated with a foot shock the US. Among 103 neurons imaged on day 2, 80.6% of CGRP$^{PBN}$ neurons responded

(average of all 10 trials) robustly after foot shock as shown before[46] (Fig. 2g, h). There was a significant increase in GCaMP responses to the light CS1 between early (1-2) and late (9-10) trials, (Supplementary Fig. 2c).

On day 3, mice were exposed to 4 trials of the tone CS2 and light CS1 spaced by 0.5 s. Consistent with previous findings[43,46], we observed that CGRP[PBN] neurons were activated by the light CS1. Out of 197 neurons imaged, 43.7% of the CGRP[PBN] neurons were activated by light CS1 and 27.4% of the neurons were activated by tone CS2 (Fig. 2i, j) with no significant difference in response to tone CS2 across the trials (Supplementary Fig. 2d). On the test day, we assessed both freezing level and fluorescent activities of CGRP[PBN] neurons in response to tone CS2. The average fluorescence of these neurons increased compared to the pre 10-s baseline during the tone CS2 presentation. Among 209 CGRP[PBN] neurons imaged on the test day, 41.2% were activated by the tone CS2, while 53.1% showed no response, and 5.7% were inhibited (Fig. 2k, l). The average area under the curve (AUC) of neurons and freezing in individual mice during tone CS2 in test phase exhibited a significant, but moderate linear correlation (Supplementary Fig. 2e). These results reveal that CGRP[PBN] neurons are innately responsive to the US and become activated in response to CS1 after FOC and to CS2 after SOC. This progression underscores CGRP[PBN] neurons' adaptability to associate both direct and indirect cues with aversive stimuli.

In the standard protocol (Fig. 1a) used for the calcium-imaging experiment described in Fig. 2, the pairing of CS1 with CS2 (day 3) occurred in the same context as paring CS1 with the US (day 2). To examine whether emergence of calcium activity after CS1 with CS2 pairing depends on the context, we repeated the experiment in Fig. 2, but on day 2 the CS1 and US pairing occurred in a novel context (Supplementary Fig. 3a). During the second-order association, both control (did not receive tone on day 3) and experimental groups showed increased neuronal activity in response to the light CS1 paired with a foot shock from the previous day (Supplementary Fig. 3b). Like the previous experiment conducted in standard protocol, the experimental group did not exhibit significant changes in response to the tone CS2 during the pairing (Supplementary Fig. 3b). On test day 4, when both groups were exposed to tone CS2, only the experimental group showed an increase in CGRP[PBN] neuronal activity in response to the tone compared to the control group (Supplementary Fig. 3c–e). In this experiment, 55 neurons (60.4%) out of 91 in the experimental group showed an increased AUC in response to tone CS2, compared to 12 neurons (18.8%) out of 64 in the control group (Supplementary Fig. 3d, e). We conclude that, like the behavioral experiment (Fig. 1f), the emergence of calcium activity in CGRP[PBN] neurons after CS1-CS2 pairing does not depend on the context.

Interestingly, we observed that the fluorescent activity was sustained for 10 s after the tone CS2 terminated during the test (Fig. 2l and Supplementary Fig. 3c). To determine whether the timing of CGRP[PBN] neuronal activity in response to the learned light CS1 was similarly sustained, we analyzed the responses to the light CS1 during FOC. The relative area under the curve (AUC) in the 10 s following light CS1, compared to the pre-stimulus baseline (10 s), remained elevated. This finding aligns with the freezing response observed during and after light CS1 (Supplementary Fig. 3f, g). The percentage of activated neurons during light CS1 and the subsequent 10 s did not show significant changes (Supplementary Fig. 3h). These results indicate that CGRP[PBN] neurons remain active even after the learned cues terminate in both FOC and SOC.

We analyzed neuronal activities of CGRP[PBN] neurons during SOC procedure by calculating AUC of traces in response to the two conditioned stimuli. While the light CS1 evoked a significant increase of average responses of all trials during and after pairing with US compared to the habituation phase (Fig. 3a, left), the tone CS2 showed increased activity after SOC rather than during learning phase, compared to the responses before learning (Fig. 3a, right). We did not see

gradual increases of responses either to the light CS1 throughout trials during CS1-US pairing (Supplementary Fig. 2c) or to the tone CS2 during CS2-CS1 pairing days (Supplementary Fig. 2d). This delayed acquisition of CS2 responsivity in CGRP[PBN] neurons during SOC compared to CS1 in FOC implies that a different network may be involved to consolidate the second-order memory.

Using K-means clustering, we identified 5 distinct categories of CGRP[PBN] neuronal activities among 209 neurons in test phase (Fig. 3b, c). CGRP[PBN] neurons displayed different peak timings within analysis window (Fig. 3c). The first cluster (31 neurons, 14.8%) exhibited a large response broadly tuned to tone CS2; they increased their activity during the tone CS2 presentation and gradually declined during the 10 s after the tone ceased. The second cluster (46 neurons, 22.0%) showed no response during the tone CS2 but started to increase after tone. The third cluster (61 neurons, 29.2%) exhibited a small, delayed increase of activity during tone CS2 then returned to baseline. The fourth cluster (55 neurons, 26.3 %) had increased activity at the onset of tone CS2 that gradually dissipated during tone presentation. The last cluster (16 neurons, 7.7%) showed no responses within a window of analysis. Similarly, clustering of CGRP[PBN] neurons showed dynamic responses during CS2-CS1 pairing phase (Supplementary Fig. 4a). Three clusters (cluster 1, 2, 5) showed robust responses to the light CS1. The cluster 3 showed a small increase to tone CS2 then a decrease during light CS1. However, during CS1-US pairing, all of clusters showed a similar pattern of strong foot shock US response (Supplementary Fig. 4b). This heterogeneity of neuronal responses in CS2-CS1 and test phases may contribute to the complex neural representation of fear-related information during SOC.

To address how the activity of individual CGRP[PBN] neurons changes across the behavioral paradigm, we traced 122 neurons across phases of the SOC procedure (habituation, CS2-CS1, and test days). Due to motion artifacts caused by the foot shocks, we were unable to include CS1-US pairing phase data in longitudinal registration process of neural activities. Within this set of neurons, we tracked activity changes in response to tone CS2 across days (Fig. 3d). Only 14.8% of neurons were activated to the tone before SOC (habituation); they exhibited a significant decrease in activity (measured as AUC) during and after learning process (Fig. 3e). Most neurons (62.3%), which were unresponsive to the tone CS2 prior to learning, showed significant increase following SOC (Fig. 3f). Another group of neurons, which showed decreased fluorescence in response to tone CS2 before SOC, showed an increase in AUC during and after the SOC procedure (Fig. 3g). These data suggest that the CGRP[PBN] neurons responsive to the CS2 were mainly derived from those that did not initially respond to the CS2 before learning.

We next examined which CGRP[PBN] neurons encode first and second-order memory. Among 122 neurons, 53 neurons (43.4%) were responsive during second-order memory recall (Fig. 3h). We asked whether they were the same neurons that were activated by first-order memory (light CS1). Of the 53 neurons responding to the tone CS2, 47.2% (25 neurons) also responded to light CS1 (CS1/CS2-encoding), the rest of them only responded to CS2 (CS2-encoding, Fig. 3h). The average trace of CS1/CS2-encoding neurons during light CS1 was greater than CS2-encoding neurons (Fig. 3i). The CS1/CS2-encoding neurons had 1.9 fold greater AUC during first-order memory recall compared to second-order memory (Fig. 3j). Also, CS1/CS2-encoding neurons displayed similar AUC to tone CS2 exposure compared to the new CS2-encoding neurons that displayed a broader but sustaining activity even after CS2 (Fig. 3k, l). These results indicate that about half of CS2 responsive CGRP[PBN] neurons respond to both first- and second-order cues and exhibited larger responses to the first-order cues than the second-order cues.

To further assess the impact of SOC on neuronal activity patterns, we calculated the maximum correlation among 122 traced neurons using Pearson's correlation test (Supplementary Fig. 4c, d).

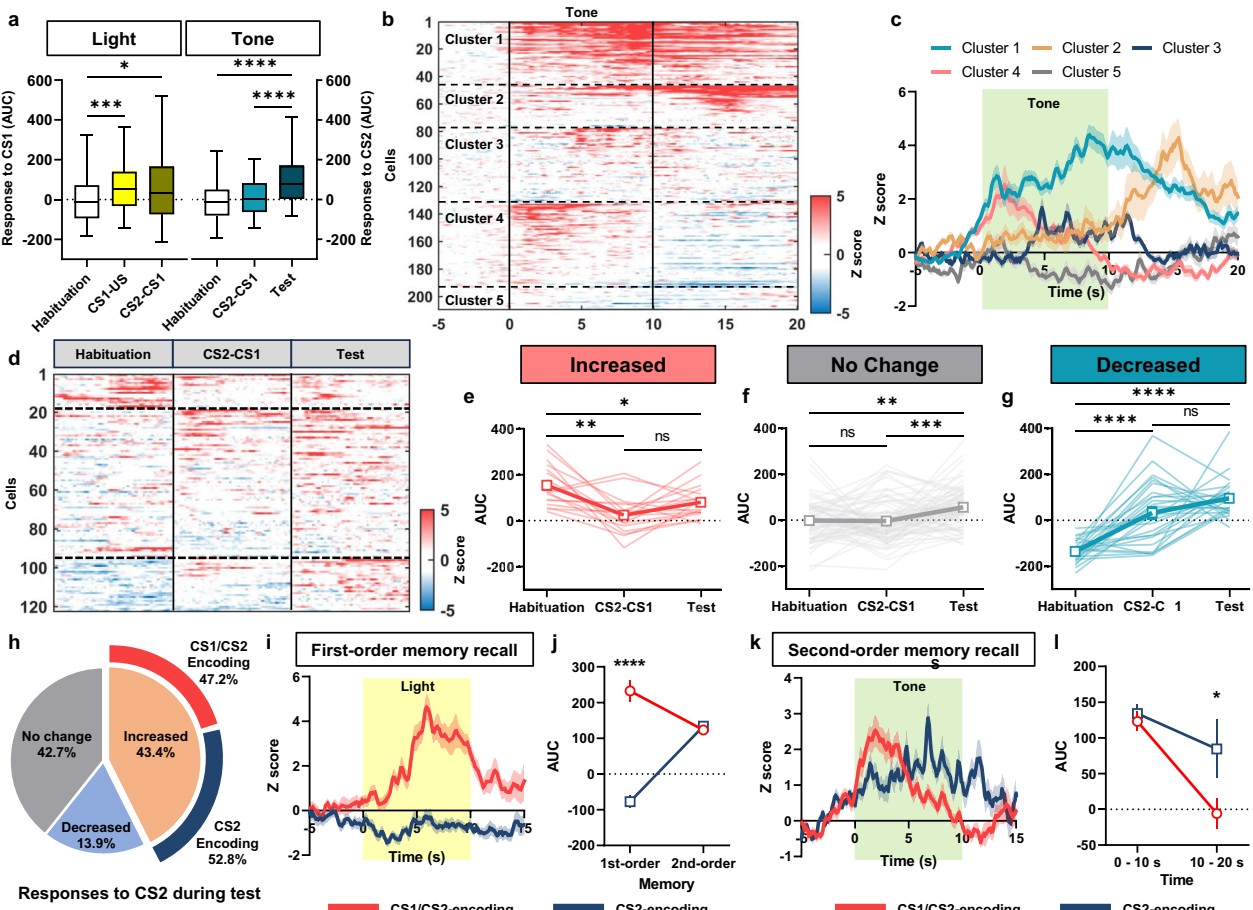

**Fig. 3 | Heterogeneity of CGRP^PBN neuronal responses to conditioned tone (CS2) after second-order conditioning. a** Average AUC of CGRP^PBN neuronal activities in response to light CS1 (left) and tone CS2 (right) during SOC ($n = 8$). The number of neurons in each day is 199 (habituation), 103 (CS1-US), 197 (CS2-CS1), and 209 (test). **b** Heatmap of clustered individual neuronal response to tone CS2 in the test phase ($n = 209$ from 8 mice). **c** K-means clustering of CGRP^PBN neurons based on their activity patterns during tone CS2 presentation on the test phase (209 neurons). **d** Heatmap of 122 neurons analyzed by longitudinal tracking of individual neuron activities to tone CS2 across experimental phases. Horizontal lines indicate classified neurons based on tone CS2 responses during habituation (top, increased; middle, no change; bottom, decreased). **e** The average changes of AUC over learning phases for neurons initially classified as an "increased" response group during habituation (18 neurons). **f** The average changes of AUC over phases for "no change" group during habituation (76 neurons). **g** The average changes of AUC over phases for "decreased" group during habituation (28 neurons). **h** Pie chart illustrating the proportion of neurons during tone CS2 in test phase and proportion of CS2 responsive neurons based on activity to light CS1. **i** Average traces of CS1/CS2 and CS2-encoding neurons during first-order memory recall (CS2-CS1 association). **j** AUC of traces during first and second-order memory recall. **k** Average traces of CS1/CS2-encoding neurons and CS2-encoding neurons during second-order memory recall. **l** AUC of traces during CS2 (0 − 10 s) and post CS2 (10 − 20 s). **i–l** $n = 25$ neurons for CS1, CS2 encoding group, $n = 28$ neurons for CS2 encoding group. Traces are average of trials. Data are mean ± SEM, *$p < 0.05$, **$p < 0.01$, ***$p < 0.001$, ****$p < 0.0001$. Source data are provided as a Source Data file.

On the habituation day, the correlations among neurons in response to tone CS2 were comparable to those observed during the baseline period (5 min before the session started) (Supplementary Fig. 4e, g). However, after the acquisition of the second-order memory, CGRP^PBN neurons exhibited a significant increase in maximum correlation in response to tone CS2 compared to the baseline (Supplementary Fig. 4f, h). These results indicate that the acquisition of SOC by a conditioned stimulus alters both the scale and coherence of neuronal activity patterns.

We hypothesized that the number of neurons responding to stimuli could reflect the aversiveness of the stimuli. Accordingly, the fewest CGRP^PBN neurons would respond to neutral cues, more would respond to noxious stimuli and learned cues, and the most neurons would respond to the foot shock US. To test this idea, we exposed mice to noxious but non-painful stimuli, such as a brief loud sound (1 s, 15 kHz, 90 dB) or bright light (1 s, 600 lux). We found that 25.6% and 21.8% of CGRP^PBN neurons, among a total of 78 neurons, were activated by the bright light and loud tone, respectively (Supplementary

Fig. 5a–e). As expected, the number of neurons responding to these stimuli was greater than the number responding to neutral cues before learning (18.1% and 14.6%, respectively) but lower than the number responding to learned cues (43.7% and 41.2%, respectively; Supplementary Fig. 5d). Additionally, 6.4% of neurons responded to both the bright light and loud tone, whereas 20.5% of neurons responded to both the light (CS1) and tone (CS2) after learning (Fig. 3h). These results reveal that distinct populations of CGRP^PBN neurons respond to the different sensory modalities and the responses of both responding populations increase after FOC and SOC.

## Inhibition of CGRP^PBN neurons during CS2-CS1 pairing attenuates second-order conditioning

To investigate the necessity of CGRP^PBN neurons in SOC, we bilaterally injected AAV-DIO-hM4Di-mCherry, an inhibitory G-protein coupled receptor[60,61], or AAV-DIO-mCherry as a control, into the PBN of *Calca*^Cre/+ mice (Fig. 4a). After allowing 4 weeks for viral expression, the mice were exposed to the habituation and CS1-US pairing phase followed by

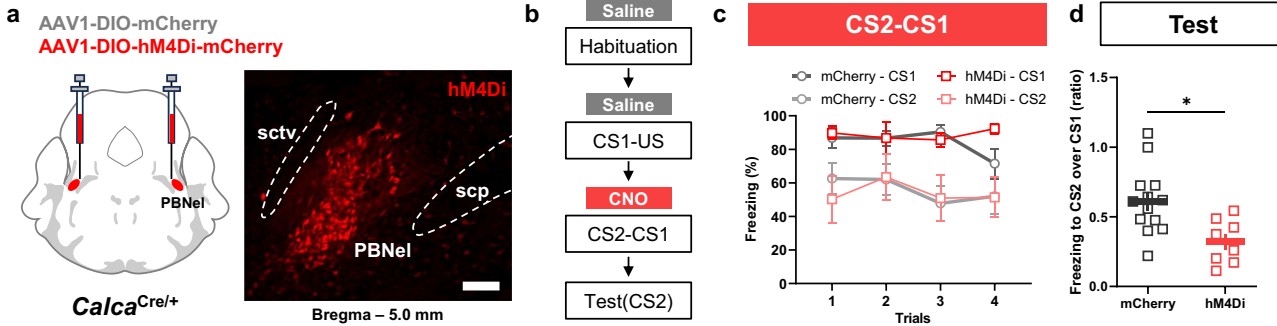

**Fig. 4 | Chemogenetic inhibition of CGRP^PBN neurons during second-order association attenuates second-order memory. a** Schematic diagram representing bilateral injection of AAV-DIO-hM4Di-mCherry (hM4Di) or AAV-DIO-mCherry (control) into PBN and representative image of viral expression in PBN. Scale bar, 100 μm. **b** Timeline summarizing experimental phases. **c** Comparison of freezing behavior during light CS1 and tone CS2 presentation in the CS2-CS1 pairing phase for hM4Di and control groups. CNO was injected 30 min prior to session ($n = 13$ for mCherry and $n = 8$ for hM4Di). **d** Second-order memory calculated as a ratio of freezing in response to tone CS2 (test) over freezing to CS1 ($n = 13$ for mCherry and $n = 8$ for hM4Di). Data are mean ± SEM, *$p < 0.05$. Source data are provided as a Source Data file.

an intraperitoneal (i.p.) injection of saline. The next day, the mice received the hM4Di ligand, Clozapine-N-Oxide (CNO, 5 mg/kg, i.p.)[62,63], 30 min before the CS2-CS1 pairing. On the test day, the freezing response to the tone CS2 was measured (Fig. 4b). The inhibition of CGRP^PBN neurons via CNO treatment did not affect freezing responses during the CS2-CS1 pairing compared to controls (Fig. 4c). However, chemogenetic inhibition of CGRP^PBN neurons during the second-order association resulted in a significant (44.2%) reduction in the freezing level to tone CS2 on the test day, compared to both the mCherry-control group (Supplementary Fig. 6a) and the experimental group (WT mice experiment in Fig. 1) trained with the standard SOC protocol (Supplementary Fig. 6b). Additionally, the hM4Di group exhibited a decreased freezing response after the 10-s tone CS2 presentation compared to the control group (Supplementary Fig. 6c). The freezing responses to the light CS1 after SOC were intact (Supplementary Fig. 6d, e). The freezing response of the hM4Di group to tone CS2 was only 31.6% of their response to light CS1, indicating a diminished strength of second-order memory compared to the 66.4% observed in the standard protocol (Fig. 4d). These results indicate that activity of CGRP^PBN neurons contributes significantly to the second-order association.

## Discussion

We established a SOC paradigm in mice employing an aversive stimulus and two distinct sensory cues. Our findings demonstrate the ability of mice to form a second-order association following FOC in the absence of the US. Through one-photon, calcium imaging of CGRP^PBN neurons in vivo, we characterized their dynamic responses throughout the learning progression. Initially, these neurons were activated exclusively by the US. However, after the CS1 was paired with the US, they began responding to CS1. While there was no increase in average response to CS2 during CS2-CS1 pairing phase, a significant 41.2% of CGRP^PBN neurons showed a strong response to CS2 on the subsequent, test day. Longitudinal tracking of neuronal activity revealed that the CS2-encoding CGRP^PBN neurons emerged from both the CS1-encoding population and neurons that were unresponsive before learning. Furthermore, chemogenetic inhibition of CGRP^PBN neurons during the second-order association phase led to a decrease in fear responses to CS2. This finding highlights the dual role of CGRP^PBN neurons: they are crucial for transmitting signals about the US and for encoding information about environmental cues that predict aversive events.

Higher-order conditioning, while not as extensively studied as FOC, likely plays a significant role in natural environments resembling the complexity of human memory formation[64]. It is well-documented that in first-order classical conditioning, the learning efficiency is maximized when the CS precedes the US[65,66]. Similarly, the present study demonstrated that successful second-order association occurs when CS2 is presented before CS1. Furthermore, we found that for SOC to be effective, the interval between CS2 and CS1 needs to be short.

We have identified several distinctive features of SOC compared to FOC. One notable difference is the weaker strength of second-order memory compared to first-order memory, evidenced by freezing levels to CS2 being around 66.4% of those elicited by CS1. This disparity likely arises from the absence of the US during CS2-CS1 pairing, leading to a less potent valence attributed to the second-order association. This feature may also contribute to faster extinction of second-order memory compared to first-order memory. Given similar extinction patterns observed in experiments with tone CS1 and light CS2 in rats[19], the faster extinction of second-order memory to tone CS2 in our study may not be due to sensory modality differences between the light and tone. The moderated strength of second-order memory might afford animals greater flexibility in their responses in a changing environment.

In our 2-day protocol of SOC, we did not observe formation of a second-order association. This finding is different than a study that reported short-term memory of CS2 following SOC in a fear-potentiated-startle study in rats[67]. This difference may be because we used freezing behavior as the measure of the conditioned response, which is an adaptive response that involves a different neural circuit than the startle reflex. Another study reported that the recall of short-term memory after FOC elicits robust activation of uncharacterized PBN neurons, suggesting that a long consolidation period is not necessary for activation of CGRP^PBN neurons by the cue after FOC[48]. Our data are compatible with their conclusion, even though we did not directly examine GCaMP responses to the cue directly after FOC as they did, because the average GCaMP response to the light during conditioning was the same as that observed the next day, there was a significant increase in GCaMP responses between first to last trials, and more CGRP^PBN neurons responded to the cue during and after FOC compared to the habituation day.

The absence of a US during the association of CS2 with CS1 can be compared to a prediction error, a fundamental concept in associative learning theory, which posits the discrepancy between real and expected outcomes as a key driver of learning[50,51,55]. Maes et al. showed that second-order memory is disrupted when midbrain dopamine neurons are inactivated at the start of the reward-predicting cue[68]. Dopamine neurons in ventral tegmental area (VTA) are known to be activated by omission of the US during fear extinction[69]. Also, dopamine neurons innervate the BLA and promote second-order fear conditioning via the dopamine D1 receptor[24]. Infusion of a NMDAR

antagonist in BLA is critical for acquisition of second-order memory when the expected shock is omitted[70]. CGRP[PBN] neurons may play a similar role in aversive learning as dopamine neurons play in appetitive learning. Activation of parabrachial neurons has been found to inhibit dopamine neurons in the VTA[71,72] and although CGRP[PBN] neurons do not project directly to the VTA, they might exert an indirect influence. Hence, CGRP[PBN] neurons could also contribute to aversive learning by inhibiting the activity of VTA dopamine neurons.

Sensory preconditioning, a variant of higher-order conditioning, was not strongly established under our test conditions, possibly due to an insufficient number of CS2-CS1 paired trials required to establish an association[10,19]. During the CS2-CS1 pairing phase, these conditioned stimuli had no inherent valance, as CS1 was associated with the US only on the subsequent day. This contrasts with SOC, where CGRP[PBN] neurons become activated during the CS2-CS1 paired trials due to the prior establishment of CS1-US association.

The calcium activity of CGRP[PBN] neurons during the learning process highlights their role in SOC. The smallest number of CGRP[PBN] neurons was activated to the neutral cues before learning and the largest number of neurons was activated by the painful US. The noxious sensory stimuli such as loud tone activated more neurons than neutral tone, but less than learned tone after SOC. Therefore, the number of neurons responding to the stimuli may represent the degree of aversion of sensory stimuli.

Given the role of CGRP[PBN] neurons as a general alarm[36], their reactivation by the CS1 could effectively generate alarm signals, acting as a surrogate US during the association with CS2. Hence, the reactivation of the US pathway by CS after learning provides insights into the neural mechanisms of higher-order conditioning. There are at least two hypotheses for what drives CGRP[PBN] neuron responses to a CS after learning. Since these neurons are known to respond to aversive sensory stimuli[36], one possibility is that signaling pathways from the CS1 to the CGRP[PBN] neurons is potentiated through conditioning process. Indeed, rabies monosynaptic tracing studies revealed that CGRP[PBN] neurons receive inputs from the canonical visual and auditory circuits[38,39], which may deliver potentiated CS1 information. Alternatively, as the association between the CS and US is consolidated, the presentation of CS might elicit physiological responses that resemble those triggered by the US (e.g., foot shock) itself[45].

Cluster analysis revealed diverse activity patterns of CGRP[PBN] neurons in response to the CS2. Three of the clusters responded to the CS2, potentially involved in encoding learned CS2 (clusters 1, 3, 4). Two clusters exhibited activity following the tone (clusters 1, 2). Considering the heterogeneity of CGRP-expressing neurons in the PBN[37,73] these distinct sub-populations likely process and encode CS and US information in diverse ways. Future studies exploring the gene expression profiles of these clusters could provide deeper insights into their functional roles and contributions to associative learning processes.

Longitudinal tracking of neurons revealed activity changes within CGRP[PBN] neurons throughout the learning process. Most of the neurons were activated in response to the US, as expected. After FOC, 43.7% of neuronal population was activated by CS1; about half of them became responsive to the tone CS2 on the test day. Because the CS1/CS2-encoding CGRP[PBN] neurons had a larger response (AUC) during CS1 than CS2, they may correlate best with the robustness of freezing behavior. Selective manipulation of this distinct population, while challenging, could yield valuable insights into their specific role in neural ensemble of first- and second-order associative memories.

Chemogenetic inhibition of CGRP[PBN] neurons during CS2-CS1 association attenuated second-order memory. Similar attenuation of first-order memory was observed in previous experiments that transiently inhibited this US pathway during foot shock[45]. The partial attenuation is presumably due to redundant US pathways mediating aversive learning, such as the amygdala, deep cerebellar nuclei, and basal forebrain[74,75]. Intriguingly, the inhibited group of mice exhibited reduced freezing levels not only during CS2 but also 10 s later. This observation aligns with the increased calcium activity detected in some CGRP[PBN] neurons both during and after CS2 exposure on the test day. Such sustained neuronal activity post-CS2 suggests that it plays a significant role in maintaining the freezing response following CS2.

Emerging evidence demonstrates the reactivation of the US pathway neurons by the CS after FOC. CGRP[PBN] neurons, for instance, are activated by the cue after both fear and taste learning[43,46]. Likewise, foot shock-responding deep cerebellar nuclei neurons projecting to PBN exhibited increased activity during re-exposure to the CS after fear conditioning[75]. In the basal forebrain, neurons expressing choline acetyltransferase (ChAT) and projecting to BLA displayed similar activity changes after learning and inhibiting them during or after learning attenuated conditioned responses to the CS[76]. CGRP[PBN] neurons project axons to the basal forebrain (nucleus basalis or substantia innominata)[37]. This suggests that CGRP[PBN] neurons could drive the activity of ChAT neurons in the basal forebrain, thereby facilitating associative learning. Furthermore, the reactivation of CGRP[PBN] neurons after learning sheds light on the neural mechanisms of higher-order conditioning, elucidating how conditioned stimuli can be associated with each other without an actual US. Unraveling the neural mechanisms of higher-order fear conditioning may provide further insights into the treatment of fear and anxiety-related disorders.

## Methods

### Animals

All experiments were approved by the University of Washington Institutional Animal Care and Use Committee and were performed under the guidelines described in the US National Institutes of Health Guide for the Care and Use of Laboratory Animals. The behavioral paradigm (Fig. 1) was established using male and female C57BL/6J mice (Jackson Lab #000664) that were 8–12 weeks old. The imaging and hM4Di inhibition experiments were performed with $Calca^{Cre/+}$ mice (Jackson Lab #33168) that were generated as described[40] and backcrossed to C57BL/6 mice for >10 generations. Another imaging session in Supplementary Fig. 3 was performed using either $Calca^{Cre/+}$ and $Calca^{frtCre/+}$::Ai162 mice (Jackson lab #031562). Expression of Cre in $Calca^{frtCre/+}$ mice requires action of FLP recombinase[43]. Stereotaxic surgeries were performed on males and females when the mice were 8-12 weeks old, and experiments began at least 4 weeks after surgery. Mice were housed on a 12-h light and dark cycle at ~22°C with food and water available ad libitum. Animals from the same litter were split randomly between control and experimental groups, with an nearly equal number of male and female mice in each group. The experiments were not powered to examine sex differences in behavior or CGRP[PBN] activity, but no differences were apparent.

### Stereotaxic surgery

Mice were anesthetized with 5% isoflurane and placed on a stereotaxic frame (Neurostar). Isoflurane was maintained 1.5– 2% during the surgical process. For calcium imaging studies in vivo, viral injections were into the external lateral region of the PBN using the coordinates; AP -4.90 mm; ML ± 1.35 mm; DV + 3.40. Viruses were injected unilaterally total 0.5 µl at 0.1 µl/min injection speed. A gradient refractive index (GRIN) lens (Inscopix) was placed above target (AP -4.80 mm; ML ± 1.7 mm; DV + 3.65). The lens was lowered at 0.1 mm/min. It included 3 stainless steel wires (0.127 mm bare, uncoated, A-M Systems) that protruded ~0.5 mm beyond the tip; this reduced motion artifacts during imaging. Lens and baseplate for the microscope was secured on the skull with super glue, C&B Metabond (Parkell) and dental cement. After 4 weeks of recovery, mice were tested to check field of view (FOV). Animals with a stable FOV were used in the experiments for the next 2-4 weeks. For chemogenetic manipulation of CGRP[PBN] neurons, the same coordinates were used for bilateral injection of AAV1-hSYN-DIO-hM4Di-mCherry (Addgene, #44362) or AAV1-DIO-mCherry

(Palmiter lab) alone as control. AAV1-Ef1a-DIO-GCaMP6m or AAV-CBA-FLPo-dsRed (Palmiter lab) was injected unilaterally (left or right side were randomly assigned). Mice were allowed to recover for 4 to 5 weeks before the start of behavioral tests. For all experiments, histological verification of targeted viral transduction was documented at the end of the experiments by histological analysis. This was especially important for the hM4Di experiments where adequate bilateral transduction was essential. Only 40% of the mice injected with virus were deemed good hits by a histologist blinded to the treatments.

## Higher-order conditioning

All procedures were performed in $28 \times 28 \times 25$ cm metal chamber connected to shock grid, controller, light, and speaker (Med Associates). For the standard SOC protocol, mice were acclimated to the metal chamber (context A) for 10 min then received 10 trials of 10-s CS1 (60 lux light), and 10 trials of CS2 (5 kHz, 70 dB tone) for 2 days. The interval between the trials (1–2 min) and order between CS1 and CS2 were randomly assigned. On the day 2, mice were placed in the context A for a 10-min baseline period, then they received 10 CS1 presentations (10 s) that co-terminated with foot shock (0.5 mA, 0.5 s) with 2-min inter-trial interval (ITI). On Day 3, mice were exposed to the context A and received CS2, followed by CS1 spaced by 0.5 s in between. This CS2 and CS1 pairing was repeated 4 times with a 2-min ITI. On day 4, the mice were placed in a chamber ($28 \times 28 \times 25$ cm) with acrylic panel walls (context B) and received 3 CS2 tones with a 2-min ITI. Annotation of freezing was measured manually using Ethovision software (Noldus). Freezing was determined frame by frame during the 10-s CS intervals as no movement of the animal. Experimenter was unaware of experimental conditions and/or genotype when freezing was measured. Percent freezing calculated as percent of time mice spend to freeze during 10 s of CS1 or CS2. A control group that was not exposed to CS2 on day 3, which resulted in no association between CS2 and CS1 and served as a background freezing level. For the experiments minimizing contextual effect (Fig. 1f, and Supplementary Fig. 3), mice received same procedure above in context B, except day 2 that procedure conducted in context A.

For the generalization of SOC experiment, the mice were presented with ten 5 and 10-kHz tones and lights during 3 days of habituation. Day 1 and 2 were the same as the standard protocol, on day 3 the mice received 4 pairings of 5-kHz tone followed 0.5 s later by light. On test day 4, mice received three 10-kHz tones then received three 5-kHz tones to allow a comparison of their responses within each animal.

For manipulation of CS2 and CS1 timing experiment, we introduced different patterns of CS2 and CS1 to each group of mice. This experiment was conducted in different context setting compared to the standard SOC protocol. To minimize contextual contributions, CS1-US association was conducted in context A, while other phases were conducted in context B. One group received CS2 and CS1 simultaneously (Simultaneous), a second group received CS1 30 s after CS2 (30-s interval), while a third group received CS1 first then CS2 with 0.5 s of interval (Reversed). Control group did not receive CS2 and Experiment group received CS2 then CS1 with 0.5 s of interval. These combinations were conducted 4 times as in the standard procedure above.

For the different procedure length experiment, a group of animals underwent the standard SOC protocol (3-days after habituation). The 2-day group received the standard procedure until CS1-US phase, then received CS2-CS1 pairing and the test with an hour apart. The 1-day group received CS1-US, CS2-CS1, and the test within the same day with an hour interval between sessions. During the hour interval, mice were in the home cage.

For extinction of second-order memory experiment, SOC control and SOC extinction groups of mice went through standard SOC procedure described above except that on day 4 (test day) the mice received 10 trials of CS2 instead of usual 3. Extinction of the CS1 response was also measured for 10 trials in different group (FOC extinction) at the next day of CS1-US.

Sensory preconditioning procedure was performed by switching day 2 and 3 of SOC procedure. We used the same parameters for shock, CS2 and CS1 for sensory preconditioning. On day 2, both control and experimental mice received CS2-CS1 association (control did not receive CS2), then the CS1-US association was on day 3. On the test day, we exposed mice 3 times to CS2 and CS1 to compare freezing response between groups and conditioned stimuli.

## Chemogenetic inhibition of CGRP$^{PBN}$ neurons

Clozapine N-oxide (CNO) was prepared in sterile saline and administered intraperitoneally (i.p.). For chemogenetic inhibition of CGRP$^{PBN}$ neurons during SOC experiment, the mice were acclimated to the i.p. injection by receiving i.p. injection of saline (10 μl/g) 30 min before the habituation session for 3 consecutive days. On day 1 saline was injected 30 min before the start of session (CS1-US association). Next day, CNO (5 mg/kg at 10 μl/g; RTI#C929) was injected to control and hM4Di group 30 min before the association between CS2 and CS1. On the test day, mice did not receive any injections prior to session. The CNO concentration was determined based on the previous literature[62,63]. The extent of viral transduction in both left and right PBN was determined histologically (see below) and only those mice with ample transduction (8 of 20 mice) on both sides (judged by someone blind to the experiment) were included in the analysis.

## Calcium imaging and analysis

After 4 weeks of recovery, mice were habituated to being handled by the experimenter. An nVista microscope (Inscopix) was attached to the baseplate once a week to check FOV. Recording proceeded with IDAS program (Inscopix) and the best e-focus was determined by visual inspection. Sometimes mice received brief air puff to see if CGRP$^{PBN}$ neurons responded to this aversive stimulus. Due to geometrical difficulties of PBN, our success rate of surgery for Ca$^{2+}$ imaging was low with 8 of 20 mice with good neuron visualization. About 26 CGRP$^{PBN}$ neurons were observed per mouse (range 15 to 44). Only mice with > 15 neurons in the FOV were used for experiments. For SOC experiments, nVista was connected to test chamber (Med Associates) via BNC cable to receive TTL (Transistor-Transistor-Logic) inputs from test chamber to match the onset and offset of calcium imaging sessions with the video recording. Stimuli onset including foot shock, tone and light were also delivered as TTL to nVista.

The imaging parameters were chosen for the least amount of photobleaching, but sufficient fluorescence (LED power, 0.2 - 0.5). Raw data were processed with IDPS software (ver. 1.9.1, Inscopix). After 4X spatial and 2X temporal down sampling, data were processed with spatial bandpass filter to reduce background noise. Then motion correction was applied to the images based on reference frame and region of interest (ROI) in FOV. Principal component analysis/Independent component analysis (PCA/ICA) in IDPS were used to extract $\Delta F/F$ of neurons from raw data. Thirteen pixel of average cell diameter, 150 of principal components, 100 ICA maximum iterations were used as parameters to identify cells using PCA/ICA[40]. In some cases, with severe motion and high background, a manual ROI analysis method was applied. The maximum projection images were used as reference of spatial information of neurons, then ROIs were manually drawn based on the boarder of neurons. All neurons were visually inspected for each cell based on shape and dynamics to verify accuracy.

For tracking of individual neurons across days were done by using longitudinal registration function of IDPS (minimum correlation, 0.5). After generating cell traces, neurons were manually inspected and removed from analysis when they lost trace some days or captured different neurons within the analysis file.

Raw data from IDPS were processed to calculate *ΔF/F* and Z score using customized MATLAB code [https://doi.org/10.5281/zenodo.10725342]. Since the output of manual ROI analysis was raw data, *ΔF/F* was calculated as $\Delta F/F = \frac{[(F-mean(F))]}{mean(F)}$. Mean (*F*) indicates average *F* of entire trace. For PCA/ICA analysis, output data were used because the output of PCA/ICA analysis is *ΔF/F*. For comparison of multiple animals, Z score was used which was calculated as $Z = \frac{[F-mean(F0)]}{Standard\ deviation(F0)}$. $F_O$ indicates *F* between -10 to 0 second. Time 0 indicates any type of stimuli onset (US, CS1, CS2).

For classification of neurons, area under the curve (AUC) of *ΔF/F* during before (-10 to 0 s) and after (0 to 10 s) events of interest were calculated. Neurons with >50% increase of AUC (after/before the event) were classified as 'increased' group of neurons. Neurons > 50% decrease of AUC were classified as 'decreased' group. The remaining neurons did not show > 50% increase or decrease were classified as a 'no change' group. K-means clustering was performed using Behavior Ensemble and Neural Trajectory Observatory (Bento) which is an open source, calcium-imaging- analysis program developed by D. Anderson and A. Kennedy[77]. Z scored traces from -10 s to +20 s of all animals were used as an input, then K-means clustering performed with group number of 5. Optimal number of clusters was determined using a Silhouette score[77].

The correlation analysis of neurons was conducted using version 23.6 of the Inscopix Data Exploration, Analysis, and Sharing (IDEAS, Inscopix) platform. Uploaded traces were analyzed using the "Compare Neural Circuit Correlations Across States" tool within IDEAS. This tool computes maximum pairwise correlations among cell traces within specific states (e.g., baseline, tone, and light). Additionally, it determines the differences in correlations between these states, represented by Cohen's d values.

## Histology and microscopy

Mice were anesthetized with Beuthansia (0.2 ml, i.p.; Merck) and transcardially perfused with phosphate-buffered saline (PBS) followed by 4% paraformaldehyde (PFA, Electron Microscopy Science) in PBS. Then brains were extracted and fixed in 4% PFA at 4°C overnight. The next day, the brains were moved to 30% sucrose for 48 hr. Brains were frozen in OCT compound (ThermoFisher) before sectioning. Coronal stions (30 μm) were made on a cryostat (Leica Microsystems), then stored in PBS. For immunostaining, 30 μm sections were washed in PBS and incubated in the blocking solution (3% normal donkey serum in PBST) for 1 hr at room temperature. Primary antibodies were applied to the solution and incubated overnight in 4 °C: chicken-anti-GFP (1:10000, Abcam, ab13970), and rabbit-anti-DsRed (1:1000, Takara, 632496). After 3 washes in PBS, sections were incubated for 1 h in PBS with secondary antibodies: Alexa Fluor 488 donkey anti-chicken (1:500, Jackson ImmunoResearch, #2340375), Alexa Fluor 594 donkey anti-rabbit (1:500, Jackson ImmunoResearch, #2340621). Then the sections were mounted onto glass slides, and coverslipped with DAPI Fluoromount-G (Southern Biotech). Sections were imaged with a Keyence BZ-X700 microscope.

## Quantification and statistical analysis

An online power and sample size calculator was used to determine an effective sample size for statistical comparisons (http://powerandsamplesize.com). Prism 9 (GraphPad) was used for all other statistical analysis. For all data, normality was tested using the Shapiro-Wilk test to determine whether parametric or non-parametric analyzes were required. For comparing two groups, Student t-test were used otherwise ANOVA (one-way/two-way or normal/repeated measured were used based on the variables) was used. The asterisks in the figures represent the *p* values of post-hoc tests corresponding to the following values *$p < 0.05$, **$p < 0.01$, ***$p < 0.001$, ****$p < 0.0001$. Following histology and imaging, any mouse whose targeted injection site

was missed or overly expressed was excluded from experimental analysis.

## Statistics and Reproducibility

Microscope image in Fig. 2a, c is the representative among 8 animals in Figs. 2 and 3. Microscope image in Fig. 4a is the representative among 8 animals in hM4Di group.

## Reporting summary

Further information on research design is available in the Nature Portfolio Reporting Summary linked to this article.

## Data availability

Source data are provided with this paper.

## Code availability

The customized code is available in GitHub [https://doi.org/10.5281/zenodo.10725342].

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

## Acknowledgements

We thank Susan Phelps and Lucy Anastas for maintaining the mouse colony, Dr. Larry Zweifel, Dr. Jeansok Kim and lab members for editorial comments, Dr. Scott Ng-Evans for device setup and Dr. Steve Thomas (University of Pennsylvania) who suggested that *Calca* neurons might mediate second-order conditioning after a seminar. This study is supported by Howard Hughes Medical Institute (HHMI).

## Author contributions

R.P. and S.P. designed research and wrote manuscripts; S.P. and A.Z. established SOC and analyzed behavioral experiments; S.P. and F.C. performed and analyzed 1-photon imaging; A.Z. generated customized code for data extraction; S.P. and F.C. analyzed Ca$^{2+}$ imaging data.

## Competing interests
The authors declare no competing interests.
