## [Peer Review File · Nature Communications]

REVIEWER COMMENTS

Reviewer #1 (Remarks to the Author):

In this manuscript, Park and colleagues report that Calca neurons in the parabrachial nucleus are critically involved in second-order conditioning (SOC) in mice. The authors successfully established a protocol for SOC using light as CS1, paired with a US (footshock), and tone as CS2. Through in vivo calcium imaging, they demonstrated that Calca neurons became responsive to CS2 after its pairing with CS1, previously associated with the US. Additionally, chemogenetic suppression of Calca neuron activity during CS1-CS2 pairing reduced freezing in response to CS2 during test sessions. The behavioral experiments were well-designed, making this study potentially interesting to a general readership, especially given the known limited robustness of SOC in mice. However, there are some conceptual questions and concerns about the manuscript in its current form.

Major points:

1. Longitudinal imaging results indicated a partial overlap between CS1- and CS2-responsive neurons, suggesting distinct populations encoding two different danger signals. In addition, results in Fig. 4 imply that Calca+ neuron activity is necessary for CS2-CS1 association on day 3. These results raise questions about the specific roles of these neurons in SOC. First, what are the physiological implications of these separate populations? Second, does the CS2-CS1 association occur in Calca+ neurons, and if so, how is CS2 information conveyed to them when the majority of Calca neurons did not respond to CS2 even during pairing (fig 3e and f)? Overall, it is not clearly discussed how the imaging data contributes to answering these critical questions.
2. To avoid pairing CS2 with the training context, it is desirable to associate CS1 and CS2 in a novel environment, as noted in the manuscript. However, all imaging experiments were conducted in the same training chamber, except for the day 4 test session. This raises the possibility of CS2-context pairing. While Fig. 1f shows CS1-CS2 association occurring in a different context, it cannot completely rule out context information contamination during CS1-CS2 association in the original training chamber during imaging sessions. This caveat should be discussed.
3. Fig 1i: The failure of the 2-day protocol to induce significant second-order memory (short-term memory at 1hr) is surprising, considering that reactivated CS1 may substitute the US during SOC. Why does SOC require a different incubation time compared to first-order conditioning? Can SOC be consolidated into long-term memory without expressing short-term memory at 1 h? Is this a general feature of SOC?
4. Fig 1f: Is the experimental group (blue) significantly different from other groups (e.g., 30-s interval)? Another caveat of this study is the small number of animals used, which is acceptable for imaging studies but not for behavioral studies shown in Fig 1.

Minor points:

1. What is the rationale for determining CNO concentration? 5 mg/kg seems much higher than usual.
2. It is interesting to note a significant amount of freezing after the tone terminated (Fig 2d). Is this specific to CS2 (tone)? Did the mice also show freezing after CS1 (light)? If so, was the Ca signal also sustained after CS1 terminated?

Reviewer #2 (Remarks to the Author):

Second-order conditioning (SOC) is an interesting phenomenon in human and model animals, including rodents, birds, fruit flies and even invertebrates. As the authors mentioned, it's surprisingly understudied in mice using modern tools, and thus lacks mechanistic insights. In this manuscript, Part et al. from Palmiter lab tried to establish SOC in mice and explored the role of CGRP neurons in PBN in SOC. Building on their previous findings of CGRP neurons in mediating first-order, the authors examined the functional role and the calcium dynamics of CGRP in SOC, suggesting the critical role of these neurons in SOC. The experiments are straightforward and well performed. The conclusions are clear and the findings are interesting.

I would only suggest one additional group of experiments to strengthen the causal role of CGRP neural response to SOC, which concerns the in vivo imaging experiments. Although the authors showed the dynamic change of CGRP neurons in SOC, there is no proper control to support the correlation between CGRP neural response and behavioral output in SOC. For example, the controls in Figure 1 (30s interval, reversed, simultaneous) will serve as excellent paradigms for in vivo imaging. While it's too demanding to ask for in vivo imaging of all three control paradigms, the authors should at least perform additional in vivo imaging using either one of these three control paradigms, to show that CGRP neurons fail to respond to CS2, which will really provide a stronger link between the CGRP neural activity and SOC behavior.

One minor typo:

Line 103, 'novel 10-kHz novel'.

Reviewer #3 (Remarks to the Author):

The manuscript by Park et al examines the role of Calca neurons in the parabrachial nucleus in the formation of associative memories. They find mice are able to form second-order associations between the original conditioned stimulus (light) and a new stimulus (tone), which gives rise to freezing in response to tone. Through Ca²⁺ imaging in vivo, they find that Calca neurons in the parabrachial nucleus respond to both the light stimulus and, after pairing, the tone stimulus. Finally they show that inhibition of Calca neurons during the light-tone pairing inhibits subsequent freezing to the tone. This is a very nice study.

Comments

1. In Figure 1C, it looks like the experimental animals show 40% freezing to the first presentation of the tone on day 3 (which occurs before the light). In Figure 1d, it looks like the control animals show ~8% freezing to the presentation of the tone on day 4. I don't understand why these numbers would be so different since both groups of mice are hearing a tone that is not (or not yet) paired to the light stimulus. Maybe I am misunderstanding the experimental design.

2. In Figure 4, chemogenetic inhibition of Calca neurons during the sound-light pairing significantly reduces subsequent freezing behavior to sound. To me, it seems that an important control is to examine the degree to which this chemogenetic inhibition affects subsequent freezing behavior to light. If chemogenetic inhibition during light-sound pairing reduces subsequent freezing to sound only (and not light), it seems very specific indeed. If it also affects subsequent freezing to light, I feel the data are somewhat harder to interpret.

3. The authors suggest that consolidation is required for first order learning because an increase in the average fluorescence of Calca neurons to light was only observed on the second day. However, Smith et al. observe strong responses to the conditioned stimulus within 10 minutes (Smith, J. A., Y. Ji, R. Lorsung, M. S. Breault, J. Koenig, N. Cramer, R. Masri and A. Keller (2023). "Parabrachial Nucleus Activity in Nociception and Pain in Awake Mice." *J Neurosci* 43(31): 5656-5667.) The authors should remove the suggestion that consolidation is required and cite this earlier work.

Minor comments

1. The manner in which the behavioral experiments were performed is not as clear as it could be. For instance, when you read the methods, it turns out that there has been three days of conditioning, not one, as suggested by the figure. In addition, it is not clear what time windows were used for the identification of freezing and how % freezing was calculated. I'm speculating that

freezing required immobility for some duration? Does 80% freezing mean that the mice froze for 8 of 10 seconds of the stimulus presentation? Please add more detail.

2. Please provide more details on the identity of Calca neurons in the PBN. E.g., Are they excitatory or inhibitory? Do they reside in a specific region of the PBN? What fraction of PBN neurons are Calca neurons? Do they include local and projection neurons? Where do they project to? I realize that this information is likely in previous publications that are cited, but the reader should not have to read all of these earlier papers to have a clear understanding of which neurons are being imaged and manipulated.

Point by point responses (NCOMMS-24-13565-T, S. Park et al.)

General remarks

We appreciate the reviewers' critical questions and constructive suggestions. We did our best to respond all the questions by conducting additional behavioral experiments, Ca²⁺ imaging and chemogenetic inhibition experiments. In response to reviewers, we added additional mice to the experiments in Fig. 1, added 2 new figures and 3 panels to existing figures. Furthermore, we revised The Introduction, Results, Discussion and Methods and refer to more papers based on reviewer's suggestions. Please see below our "point by point responses" for the details. Our responses to questions and all the changes in manuscript are colored blue.

Reviewer #1 (Remarks to the Author):

In this manuscript, Park and colleagues report that Calca neurons in the parabrachial nucleus are critically involved in second-order conditioning (SOC) in mice. The authors successfully established a protocol for SOC using light as CS1, paired with a US (foot shock), and tone as CS2. Through in vivo calcium imaging, they demonstrated that Calca neurons became responsive to CS2 after its pairing with CS1, previously associated with the US. Additionally, chemogenetic suppression of Calca neuron activity during CS1-CS2 pairing reduced freezing in response to CS2 during test sessions. The behavioral experiments were well-designed, making this study potentially interesting to a general readership, especially given the known limited robustness of SOC in mice. However, there are some conceptual questions and concerns about the manuscript in its current form.

Major points:

1. Longitudinal imaging results indicated a partial overlap between CS1- and CS2-responsive neurons, suggesting distinct populations encoding two different danger signals. In addition, results in Fig. 4 imply that Calca+ neuron activity is necessary for CS2-CS1 association on day 3. These results raise questions about the specific roles of these neurons in SOC.

First, **what are the physiological implications of these separate populations?**

Thank you for the insightful comment. This is indeed a fundamental question that we are eager to explore further. CGRP^{PBN} neurons are known to respond to threats from various sensory modalities, including visual and auditory stimuli (S. Kang et al., 2022, PMID: 35977501). To investigate their physiological role in SOC, we asked whether the same neurons are activated by noxious tone or light. We introduced a brief loud tone (15 kHz, 90 dB) or a bright light (600 lux), which may be noxious initially but not painful, to the mice and imaged CGRP^{PBN} neuronal responses (**Extended Data Fig. 5a-c**). We observed that 21.8% and 25.6% of neurons were activated by the noxious tone and light, respectively, among 78 neurons from 4 mice (**Extended Data Fig. 5d**). Only 5 neurons (6.4%) responded to both the tone and light which suggests that different subpopulation CGRP^{PBN} neurons respond to these stimuli, like the findings S. Kang et al, (2022, PMID: 35977501). In our SOC experiment (Fig. 2), 14.7 % and 18 % of neurons were activated to the neutral tone CS2 and light CS1 before learning and 41.4% and 43.7% of neurons responded, respectively, after learning. A greater

proportion of neurons became responsive to both stimuli (20.5%) following SOC perhaps because of the association with the painful foot shock.

Please see **Extended Data Figure 5 and lines 269-281 and 362-366**.

Second, does the CS2-CS1 association occur in Calca+ neurons, and if so, **how is CS2 information conveyed to them when the majority of Calca neurons did not respond to CS2 even during pairing** (fig 3e and f)?

We appreciate the reviewer's question about the mechanisms of how the tone CS2 is conveyed to the CGRP^{PBN} neurons. Given that CGRP^{PBN} neurons are activated by various sensory modalities, including tactile, visual, auditory, and gustatory signals (C. Campos et al., 2018, PMID: 29562230; S. Kang et al., 2022, PMID: 35977501; L. Condon et al., 2024, PMID: 38583149), it is likely that they receive inputs from brain regions responsible for processing each sensory modality. Monosynaptic rabies tracing experiments (S. Kang et al., 2022, PMID: 35977501; M. Korkutata, et al., 2024, PMID: 38766214) suggest that the superior colliculus may deliver visual information about light CS1 and the inferior colliculus, a canonical auditory pathway, could potentially relay auditory information such as tone CS2. Alternatively, the learned tone CS2 might be associated with US in the basolateral amygdala (BLA) and then reach the PBN via the central amygdala (CEA).

Based on the observation that CGRP^{PBN} neuronal responses during CS2-CS1 pairing do not change dramatically, the association may occur in other brain areas such as BLA and CEA during pairing. After the consolidation of memory, when the CS becomes aversive, these neurons could be activated by the CS. This is consistent with our other observations showing that neutral CS before learning activates the least number of CGRP^{PBN} neurons, but much larger population become activated after learning.

Since we recognize the importance of this question for understanding the mechanisms of SOC, we have added a discussion on this topic in the revised manuscript. Additionally, we have clarified that the data do not suggest that the CS2-CS1 association necessarily occurs within CGRP^{PBN} neurons.

Please see **line 373-375**.

Overall, it is not clearly discussed how the imaging data contributes to answering these critical questions.

Thank you for the comment. We edited discussion describing Ca²⁺ imaging to answer reviewer's questions.

2. To avoid pairing CS2 with the training context, it is desirable to associate CS1 and CS2 in a novel environment, as noted in the manuscript. **However, all imaging experiments were conducted in the same training chamber, except for the day 4 test session.** This raises the possibility of CS2-context pairing. While Fig. 1f shows CS1-CS2 association occurring in a different context, it cannot completely rule out context information contamination during CS1-CS2 association in the original training

chamber during imaging sessions. This caveat should be discussed.

Thank you for your comments. As the reviewer pointed out, our Ca²⁺ imaging experimental results could not entirely rule out the contextual contribution to SOC, even though we observed context did not contribute to SOC in wild-type (WT) animals (Fig. 1f). To address this concern, we conducted an additional Ca²⁺ imaging experiment following the same procedure as described in Fig. 1f, using a novel context during CS1-CS2 pairing.

We have detailed the results in **Extended Data Fig. 3**. In this experiment, approximately 156 neurons from control (no second order association, $n = 4$) and experimental animals ($n = 4$) displayed a similar activity pattern to those observed in Fig. 2. Experimental animals exhibited similar patterns to the tone CS2 and light CS1 throughout SOC paradigm. The control animals showed increased activity to the light CS1 after FOC but did not show robust activity to the tone CS2 after SOC.

These results indicate that the tone CS2 was indeed associated with the light CS1 in both experimental paradigms, consistent with the WT animal experiments. Furthermore, the comparison with control animals suggests that the increased activity of CGRP^{PBN} neurons in response to the tone CS2, as compared to the control group, is mediated by second-order association. We are grateful for the reviewer's comment, which prompted us to perform additional experiments and confirm that contextual contribution to the SOC is minimal in either protocol.

Please see **Extended Data Fig. 3a-e** and **line 190-213**.

3. Fig 1i: The failure of the 2-day protocol to induce significant second-order memory (short-term memory at 1hr) is surprising, considering that reactivated CS1 may substitute the US during SOC. **Why does SOC require a different incubation time compared to first-order conditioning?** Can SOC be consolidated into long-term memory without expressing short-term memory at 1 h? Is this a general feature of SOC?

We greatly appreciate this critical comment. We had not considered this point in the previous manuscript. There is limited literature examining short-term memory after SOC, but a study by Jonathan C. Gewirtz and Michael Davis in 1997 (PMID: 9242405) demonstrates that rats exhibit a fear-potentiated startle response to the learned CS2 after SOC, which suggests the behavioral response related to short term memory. Therefore, SOC can induce short-term memory-related responses under certain experimental conditions.

There are several potential reasons why we did not observe a short-term memory response in our SOC paradigm. We used freezing as the indicator of the conditioned response rather than the startle reflex. Freezing is an adaptive response that may recruit different neural circuits than the startle reflex, it might not be as sensitive to short-term memory. Second, the insufficient pairing and weaker stimulus intensity in our study may have influenced the results. To minimize the extinction of first-order memory, we used only four pairings between CS2 and CS1, which may not have been sufficient to elicit a solid freezing response after one hour of second order association

We added a paragraph addressing the reviewer's question and our explanations in the Discussion section of the revised manuscript.

Please see the **line 333-343**.

4. Fig 1f: Is the experimental group (blue) significantly different from other groups (e.g., 30-s interval)? **Another caveat of this study is the small number of animals used**, which is acceptable for imaging studies but not for behavioral studies shown in Fig 1.

Thank you for your comment. The experimental group demonstrated statistically significant (as described in summary of the statistical analysis table in manuscript) differences compared to all other groups in Fig. 1f. In response to the reviewer's concern, we have increased the number of animals not only in this experiment but also in other experiments presented in Fig. 1. The additional animals yielded results consistent with those of the initial group. Please refer to the table below for the updated number of animals used in these experiments. All the statistical information is in the **Summary of the statistical analysis** section in manuscript.

Figure	Previous number	Current number
Fig 1d	5 for control 5 for experiment	14 for control 15 for experiment
Fig 1e	5 experiment	9 for experiment
Fig 1f	5 for control 5 for experiment 5 for 30-s interval 5 for reversed 5 for simultaneous	12 for control 13 for experiment 10 for 30-s interval 11 for reversed 11 for simultaneous
Fig 1i	5 for 3-day 6 for 2-day 6 for 1-day	9 for 3-day 10 for 2-day 10 for 1-day 5 for control
Extended data Fig 1c	5 for control 11 for SOC extinction	9 for control 15 for SOC extinction

Table 1. Number of animals used in Figure 1.

Minor points:

1. What is the rationale for determining CNO concentration? 5 mg/kg seems much higher than usual.

Thank you for pointing this out. Our rationale for the concentration used is based on the persistent inhibition throughout and after second-order association. Based on other studies (H. Zhou et al., 2023, PMID: 36690899; S. Singh et al., 2022, PMID: 36269044), we chose to use a similar concentration to inhibit CGRP^{PBN} neurons. For inhibition studies, authors sometimes use higher concentrations of CNO to make sure it is effective and lasts longer than lower doses. We did not observe any behavioral changes in the control (mCherry) animals during treatment. We have added citations of the above-mentioned papers in the Methods section.

Please see **line 287-289 and 701**.

2. It is interesting to note a significant amount of freezing after the tone terminated (Fig 2d). Is this specific to CS2 (tone)? **Did the mice also show freezing after CS1 (light)? If so, was the Ca signal also sustained after CS1 terminated?**

Thank you for the interesting question. We analyzed the freezing level of animals in response to the light CS1 during second order association (**Extended Data Fig. 3f-h**). The animals used in Ca²⁺ imaging experiment exhibited robust freezing (70.3 % in average) to the light CS1 during the second-order association session. These animals also exhibited freezing even during post 10 s after light CS1 exposure (62.9 % in average), like the behavioral response to tone CS2 after SOC (Fig. 2d). The neural activity of the CGRP^{PBN} neurons after light CS1 is consistent with the behavioral responses. These neurons exhibited increased AUC during CS1 exposure and post 10 s after CS1 compared to the pre 10 s of CS1, although post 10 s AUC is significantly lower than the AUC during CS1. Therefore, these neurons exhibit sustained activity after aversive stimuli which is consistent to the previous studies showing increased activity for about 10 s after the termination of aversive stimuli (C. Campos et al., 2018, PMID: 29562230; S. Kang et al., 2022, PMID: 35977501).

Please see **Extended Data Figure 3f-h and line 205-213**.

Reviewer #2 (Remarks to the Author):

Second-order conditioning (SOC) is an interesting phenomenon in human and model animals, including rodents, birds, fruit flies and even invertebrates. As the authors mentioned, it's surprisingly understudied in mice using modern tools, and thus lacks mechanistic insights. In this manuscript, Part et al. from Palmiter lab tried to establish SOC in mice and explored the role of CGRP neurons in PBN in SOC. Building on their previous findings of CGRP neurons in mediating first-order, the authors examined the functional role and the calcium dynamics of CGRP in SOC, suggesting the critical role of these neurons in SOC. The experiments are straightforward and well performed. The conclusions are clear and the findings are interesting.

I would only suggest one additional group of experiments to strengthen the causal role of CGRP neural response to SOC, which concerns the in vivo imaging experiments. Although the authors showed the dynamic change of CGRP neurons in SOC, there is no proper control to support the correlation between CGRP neural response and behavioral output in SOC. For example, the controls in Figure 1 (30s interval, reversed, simultaneous) will serve as excellent paradigms for in vivo imaging. While it's too demanding to ask for in vivo imaging of all three control paradigms, the authors should at least perform additional in vivo imaging using either one of these three control paradigms, to show that CGRP neurons fail to respond to CS2, which will really provide a stronger link between the CGRP neural activity and SOC behavior.

Thank you for the constructive comment. Following the reviewer's suggestion, we conducted an additional Ca²⁺ imaging experiments with control animals that underwent first-order conditioning (FOC) but not second-order association (**Extended Data Fig. 3**). These control animals exhibited a robust response to the light CS1, but no response to the tone CS2 on the test day as we expected.

Please see **Extended Data Figure 3 and line 190-213**.

One minor typo:

Line 103, 'novel 10-kHz novel'.

Thank you for pointing out this error. We fixed this typo.

Reviewer #3 (Remarks to the Author):

The manuscript by Park et al examines the role of Calca neurons in the parabrachial nucleus in the formation of associative memories. They find mice are able to form second-order associations between the original conditioned stimulus (light) and a new stimulus (tone), which gives rise to freezing in response to tone. Through Ca²⁺ imaging in vivo, they find that Calca neurons in the parabrachial nucleus respond to both the light stimulus and, after pairing, the tone stimulus. Finally they show that inhibition of Calca neurons during the light-tone pairing inhibits subsequent freezing to the tone. This is a very nice study.

Comments

1. In Figure 1C, it looks like the experimental animals show 40% freezing to the first presentation of the tone on day 3 (which occurs before the light). In Figure 1d, it looks like the control animals show ~8% freezing to the presentation of the tone on day 4. **I don't understand why these numbers would be so different since both groups of mice are hearing a tone that is not (or not yet) paired to the light stimulus.** Maybe I am misunderstanding the experimental design.

We apologize for the confusion. The experimental animals showed approximately 40% freezing on day 3 because they were in the same context where they received the foot shock on day 2. Therefore, the higher freezing response to the tone on day 3 is presumably due to contextual memory. Thanks to the reviewer's comment, we have revised the figure to reduce any potential confusion.

Figure 1. Before revise (left) and after revised (right) version.

2. In Figure 4, chemogenetic inhibition of Calca neurons during the sound-light pairing significantly reduces subsequent freezing behavior to sound. To me, **it seems that an important control is to examine the degree to which this chemogenetic inhibition affects subsequent freezing behavior to light.** If chemogenetic inhibition during light-sound pairing reduces subsequent freezing to sound

only (and not light), it seems very specific indeed. If it also affects subsequent freezing to light, I feel the data are somewhat harder to interpret.

Thank you for the valuable comment. To address this question, we conducted an experiment where we administered CNO during SOC and then exposed the mice to the light CS1 the following day without CNO (**Extended Data Fig. 6d, e**). Both the control and hM4Di animals exhibited similarly high levels of freezing to the light CS1 after SOC as much as we have seen in previous experiment. These results suggest that transient inhibition of CGRP^{PBN} neurons after first-order conditioning does not attenuate first-order memory. We added these new data in our revised manuscript.

Please see **Extended Data Fig. 6d, e** and **line 296-297**.

3. The authors suggest that consolidation is required for first-order learning because an increase in the average fluorescence of Calca neurons to light was only observed on the second day. However, Smith et al. observe strong responses to the conditioned stimulus within 10 minutes (Smith, J. A., Y. Ji, R. Lorsung, M. S. Breault, J. Koenig, N. Cramer, R. Masri and A. Keller (2023). "Parabrachial Nucleus Activity in Nociception and Pain in Awake Mice." *J Neurosci* 43(31): 5656-5667.) **The authors should remove the suggestion that consolidation is required and cite this earlier work.**

We regret the confusion on this issue. The average activity GCaMP activity of CGRP neurons to the cue during the 10 CS1-US training trials (Fig. 2g) is the same that observed the next day measured as area under the curve (Fig. 3a). In addition, the number of CGRP neurons activated during training increased (Fig. 2h) compared to the habituation day and the GCaMP activity of the CGRP neurons during the last 2 trials on training day was significantly higher than during the first 2 trials (**Extended data Fig. 2c**). Thus, FOC does not require consolidation. We make this point more explicitly in the revised text and cite the Smith et al (ref #48, PMID: 37451980). It is difficult to compare the magnitude and time course of the effects in our study and the Smith et al. paper because the parameters of the two studies are quite different.

Please see **lines 214-222** and **333-343**.

Minor comments

1. The manner in which the behavioral experiments were performed is not as clear as it could be.

For instance, when you read the methods, it turns out that there has been three days of conditioning, not one, as suggested by the figure. In addition, it is not clear what time windows were used for the identification of freezing and how % freezing was calculated. I'm speculating that freezing required immobility for some duration? Does 80% freezing mean that the mice froze for 8 of 10 seconds of the stimulus presentation? Please add more detail.

Thank you for pointing this out. The freezing is measured for 10 seconds of analysis window during stimuli, then calculated as % based on the time mice spent freezing within analysis window. We have revised the protocol in the Methods section to describe the procedures more clearly.

Please see the **line 649-661**.

2. **Please provide more details on the identity of Calca neurons in the PBN.** E.g., Are they excitatory or inhibitory? Do they reside in a specific region of the PBN? What fraction of PBN neurons are Calca neurons? Do they include local and projection neurons? Where do they project to? I realize that this information is likely in previous publications that are cited, but the reader should not have to read all of these earlier papers to have a clear understanding of which neurons are being imaged and manipulated.

Thank you for the comment. We have added more details about CGRP^{PBN} neurons in the Introduction section to better characterize these neurons for general readers. Please see below revised paragraph.

Please see the **line 56-60**.

3. Maybe I missed this, but **please confirm that the manual identification of freezing episodes was performed blind to treatment group** (or, if not, please re-quantify in a blinded manner).

Yes. Data were analyzed in a blinded manner. Experimenter was unaware of experimental conditions and/or genotype when freezing was measured. We mentioned that in the Method section.

Please see the **line 660-661**.

REVIEWERS' COMMENTS

Reviewer #1 (Remarks to the Author):

The authors addressed all of my questions and concerns. Particularly, it is great to see additional experimental data shown in Extended Figure 3. The new data and discussion are impressive and this reviewer recommends the acceptance of the manuscript in Nature Communications!

Reviewer #2 (Remarks to the Author):

thank you for the revised manuscript and my concerns are addressed.

Reviewer #3 (Remarks to the Author):

The reviewers have now addressed my concerns.

REVIEWERS' COMMENTS

Reviewer #1 (Remarks to the Author):

The authors addressed all of my questions and concerns. Particularly, it is great to see additional experimental data shown in Extended Figure 3. The new data and discussion are impressive and this reviewer recommends the acceptance of the manuscript in Nature Communications!

- Thank you very much for the positive evaluation of our manuscript.

Reviewer #2 (Remarks to the Author):

thank you for the revised manuscript and my concerns are addressed.

- We are glad that we have addressed the reviewer's concerns.

Reviewer #3 (Remarks to the Author):

The reviewers have now addressed my concerns.

- Thank you for the valuable comments; we are happy that we have resolved your concerns.